# Extra-Neural Metastases of Late Recurrent Myxopapillary Ependymoma to Left Lumbar Paravertebral Muscles: Case Report and Review of the Literature

**DOI:** 10.3390/brainsci12091227

**Published:** 2022-09-10

**Authors:** Ciro Mastantuoni, Fabio Tortora, Roberto Tafuto, Mario Tortora, Francesco Briganti, Raduan Ahmed Franca, Rosa Della Monica, Mariella Cuomo, Lorenzo Chiariotti, Felice Esposito, Teresa Somma

**Affiliations:** 1Department of Neuroscience and Reproductive and Dental Sciences, Division of Neurosurgery, A.O.U. Federico II, 80131 Naples, Italy; 2Department of Advanced Biomedical Science, A.O.U. Federico II, 80131 Naples, Italy; 3Department of Molecular Medicine and Medical Biotechnology, A.O.U. Federico II, 80131 Naples, Italy; 4Ceinge Biotecnologie Avanzate Scarl, 80131 Naples, Italy

**Keywords:** myxopapillary ependymoma, metastases, neuro-oncology, spine, molecular features

## Abstract

Ependymomas are commonly classified as low-grade tumors, although they may harbor a malignant behavior characterized by distant neural dissemination and spinal drop metastasis. Extra-CNS ependymoma metastases are extremely rare and only few cases have been reported in the lung, lymph nodes, pleura, mediastinum, liver, bone, and diaphragmatic, abdominal, and pelvic muscles. A review of the literature yielded 14 other case reports metastasizing outside the central nervous system, but to our knowledge, no studies describe metastasis in the paravertebral muscles. Herein, we report the case of a 39-year-old patient with a paraspinal muscles metastasis from a myxopapillary ependymoma. The neoplasm was surgically excised and histologically and molecularly analyzed. Both the analyses were consistent with the diagnosis of muscle metastases of myxopapillary ependymoma. The here-presented case report is first case in the literature of a paraspinal muscles metastasis of myxopapillary ependymoma.

## 1. Introduction

Ependymomas are the most common primary tumors of the spinal cord. They can arise: (1) from the ependymal cells lining the ventricles of the brain and the central canal of the spinal cord; (2) from clusters of ependymal cells in the filum terminale; and (3) from ependymal relics of embryonic development. Ependymomas rarely metastasize outside the central nervous system (CNS) [1]. The prognosis is poor and often ependymomas recur [2,3]. They are subdivided into sub-ependymomas, anaplastic ependymomas and myxopapillary ependymomas [4]. The myxopapillary ependymomas (MPEs) were described for the first time in 1932 [5], and account for approximately 27% of spinal ependymomas. They are mostly located at the level of the conus medullaris or cauda equina [6]. MPE is classified as a low-grade tumor [7] with a slow clinical course and has been histologically classified as grade 2, according to the World Health Organization (WHO) classification. Nevertheless, it may harbor a malignant behavior characterized by distant neural dissemination and spinal drop metastasis. The clinical presentation is subtle and non-specific, thus most patients experience mild symptoms for months or years before diagnosis. Total surgical resection is the mainstay of the treatment. According to European Association of Neuro-Oncology (EANO) guidelines, in cases of incomplete excision, local radiotherapy can be used as an effective adjuvant treatment to control the diffusion of tumor cells, although its ultimate role and the optimal dose is still controversial [8,9]. Ependymoma extra neural metastasis are extremely rare. Only 14 cases of intradural ependymomas metastasizing outside the central nervous system have been reported since 1955 [10]. Thus, the aim of the here-presented study is to disclose our experience with a paraspinal muscles metastasis from a myxopapillary ependymoma, reviewing the current literature in order to shed light on its differential diagnosis and treatment strategy.

## 2. Materials and Methods

Clinical, imaging, and histopathologic data were obtained in our institution. Surgery was performed at the Division of Neurosurgery of the University Hospital Federico II, Naples. Molecular characterization of the tumor was conducted at CEINGE, Advanced Biotechnologies, Naples, Italy. Written informed consent was obtained from the patient after discharge.

The surgical sample for the pathological exam was fixed in formalin overnight. Histological sections were obtained from formalin-fixed paraffin-embedded (FFPE) tissue and were stained with hematoxylin and eosin staining. Immunohistochemistry was performed for glial fibrillary acid protein (GFAP), S100 protein, cytokeratin-pan (CKAE1/AE3), cytokeratin 5/6, epithelial membrane antigen (EMA), p40 protein, and Ki67 proliferation index.

The review of the literature was conducted on PubMed using the research terms: “Ependymomas” AND “metastasis” AND/OR “Ependymomas metastasis” AND/OR “extraneural ependymoma metastasis”. The search was limited to articles published from January 1950 to January 2022 in the English language only. All studies were selected based on the following inclusion criteria: (1) series reporting extraneural ependymoma metastasis and (2) case reports reporting extraneural ependymoma metastasis. Exclusion criteria were the following: (1) studies published in languages other than English with no available English translation and (2) studies with overlapping patient populations; in this latter circumstance, the most recent series was included. All papers were screened by title, abstract, and main text by three senior authors (T.S., F.T., and F.E.).

## 3. Results

### 3.1. Case History

A 39-year-old patient came to our attention due to severe and progressive lumbar pain and voluminous left paravertebral swelling at the level L2–L3, occurring three months before. Such symptoms were refractory to medical therapy with non-steroidal anti-inflammatory drugs (NSAIDs) and opioids. His previous medical records reported a history of complex neuro-cutaneous syndrome with multiple cerebral cavernous malformations and cauda equina spinal myxopapillary ependymoma extending from L2 to L4, firstly surgically removed at age of 9 years old. Subsequently, he experienced two loco-regional relapses, five and nine years after surgery, managed with subtotal resection and multiple cycles of local radiotherapy. Thereafter, the patient had a full recovery and remained free of symptoms for 22 years. During this period, magnetic resonance imaging (MRI) follow-up exams detected nodular T2-hyperintense areas in the optic tract, in the endoventricular site, and in the spinal cord at level of D9, D10, and L1-S2 tract.

Upon admission in our institution, the neurological examination was negative. The ultrasound unveiled a hypo-anechoic formation with polylobate margins and a sub-fascial extension to a deep muscle site (maximum diameter: 38 mm), without relevant vascular signals on color-Doppler and power Doppler (PD) (Figure 1). The lumbo-sacral MRI examination showed in the left para-vertebral muscle, in correspondence of the level L2-L3, a grossly oval tissue with a diameter of 50 × 30 × 23 mm (cranio-caudal diameter (CCd) × antero-posterior diameter (APd) × latero-lateral diameter (LLd)). It had uneven signal due to the presence of fluid-fluid levels and hemoglobin degradation products in various states of evolution and inhomogeneous post-contrast enhancement (Figure 2). Furthermore, a lesion extending from L4 to S2 (maximum diameters:14 × 57 mm) with calcified and intra-lesioned hemorrhagic components was detected along with a thickening of the roots of the cauda equina and a pseudo-nodular lesion at L4 (Figure 3).

### 3.2. Surgical Approach

Our spinal board opted for surgical removal of the paravertebral mass, aimed at relieving lumbar pain, that accounted for the sole symptom, and histologically characterizing the lesion. Therefore, after informed consent, under general anesthesia and using intraoperative fluoroscopy, the patient was placed in a prone position, with a supporting roll under each iliac spine. A paramedian vertical midline incision was performed, exposing the muscular fascia. After dissection of the paraspinal muscles, a 4 cm hard-elastic reddish mass was visualized and removed en bloc (Figure 4).

The postoperative course was free of neurological complications and the patient was successfully mobilized on the second post-op day and discharged on the fifth day. Nine months after surgery, he was free of recurrences and neurologically intact.

### 3.3. Pathological Features

Macroscopic examination revealed a 50 × 30 × 25 mm mass, brown-to-red, with hard-elastic consistency. On cut section, a whitish, sharply outlined, multilobulated nodule measuring 30 × 25 × 24 mm, harder than surrounding tissue, was visible. Microscopic examination (Figure 5) showed a well capsulated tumor having a biphasic growth pattern characterized by more cellular areas alternating with looser ones. The hypercellular component was composed of epithelioid cells exhibiting slight cytologic atypia, arranged in a diffuse pseudo-papillary architecture with interposed vascular lacunae. Moreover, looser areas appeared microcystic due to intercellular deposition of mucoid material. Micro-foci of necrosis and occasional mitotic figures were evident, but no atypical mitoses were detected. Immunohistochemically, the tumor showed widespread positivity for glial fibrillary acidic protein (GFAP) and S100 protein, negativity to cytokeratin-pan, cytokeratin 5/6, epithelial membrane antigen (EMA) and p40, and a cellular proliferation index Ki67 < 10% (Table 1). These histological and immunohistochemical findings were consistent with a muscular localization of the myxopapillary ependymoma, with the same features previously diagnosed.

### 3.4. Epigenetic and Molecular Features

Epigenomic characterization of tumor lesion was performed using Infinium MethylationEPIC BeadsChip Kit (Illumina Technologies, San Diego, CA, USA). Raw IDAT files were analyzed by bioinformatic tool previously published (DKFZ, https://www.dkfz.de/de/index.html, brain classifier_vb11; version 3.1, accessed the 3 March 2022). The results of the analysis confirm the histopathological diagnosis of MPE with a “calibrated score” = 0.97. The match to a specific methylation class is confident when >0.8 [11].

Copy Number Variations (CNVs) analyses showed different chromosomal aberrations that, however, were not sufficient to discriminate between tumorigenesis or tumor metastasis. In particular, we found amplification of chromosome 9, with strong amplification of CDKN2A/B and PTCH1; amplification with gain of materials on chromosomes 7-9-11-13-17-18-20-21. (Figure 6). In particular, the gain of chromosome 11, 13, 17, 20, and 21 identified in this case are highly specific for MPE [12,13,14].

Then, we analyzed the epigenomic profile of the tumors based on a dedicated algorithm provided by DKFZ that allowed us to rapidly compare a diagnostic case with over 2800 cases of a reference cohort obtained from TGCA bank. The sample showed a strong match with methylation class of “ependymoma, myxopapillary”.

## 4. Discussion

Ependymomas are rare neuroepithelial neoplasms accounting for 1.9% of primary CNS tumors and up to 60% of primary tumors of the spinal cord, in adults. MPEs represent almost 27% of all ependymomas and are primarily intradural extramedullary tumors located more often in the lumbar thecal sac in proximity to the conus medullaris, cauda equina or filum terminale (intraspinal variant) [7]. A few common localizations are the pre- or post-sacral regions, in the so-called extraspinal variant. They generally arise in the 4th decade of life and only 8–20% occur in children [15,16]. The World Health Organization (WHO) Classification of Tumor of the Central Nervous System stated that myxopapillary ependymoma typically lacks histopathological signs of malignancy, classifying them as Grade II [7]. Although they have a benign course, MPEs may recur locally and metastasize, via both CSF (to other CNS sites) and blood (to extra-CNS locations), even in early stages of the disease. The tendency to recurrence and spreading is strongly related to incomplete resection and capsule violation [17]. Approximately half of the patients with ependymoma will experience a recurrent disease and the prognosis for these patients is poor [8,18,19,20,21]. Nevertheless, spinal cord ependymomas have a better prognosis than spinal cord astrocytomas, but factors affecting prognosis have not been defined except for gross total resection (GTR) [22].

Indeed, the gold standard treatment for primary spinal ependymoma is an early complete resection, since good functional outcomes are related to small tumor size and good neurological status at the time of surgery. Postoperative local radiotherapy is usually deployed when GTR cannot be safely achieved as per tumor features, namely infiltration of spinal cord or nerve roots, irregular shape, and production of a myxoid matrix (particularly in the filum terminale) or location [8,23]. Contrariwise, some reports support the brunt of radiotherapy also following GTR in determining better local control in both adult and pediatric population with MPE [24,25,26,27]. This is a crucial difference between MPE and the other spinal ependymoma reflecting the local aggressiveness of the former, also related to the difficulty in achieving GTR and strong correlation between capsular violation at surgery and recurrence [28].

Chemotherapy with etoposide proved to be effective in some preliminary studies and it is currently employed for recurrent cases [29].

It is only within the last 60 years that central nervous system tumors with distant metastases have been described, although they remain exceedingly rare. A fortiori, only a few reports of different kinds of extra neural metastasizing CNS ependymomas can be revealed by a survey of the literature. Due to their rarity, information about the incidence of extra neural metastasizing spinal cord tumors is not available. Fourteen cases have been reported in literature since 1955. The relevant features and the localization of the metastasis of published cases are summarized in Table 2.

The secondary lesions were localized in most cases in the lungs, pleura, lymph nodes, and liver. In almost all the symptomatic cases, the main treatment was the total resection of the lesion, but, in the few asymptomatic patients, a strategy based upon close clinical and MRI follow-up was adopted. To the best of our knowledge, this is the first case in the literature of a paraspinal muscles metastasis of myxopapillary ependymoma. Its prompt diagnosis may be hindered since it mimics several muscular primary and secondary lesions such as lipomatous lesions, sarcomas, rhabdomyosarcomas, intramuscular myxomas, and extra neural metastasis [30,31,32]. Radiological features of these lesions are reported in Table 3 [33].

As compared to the aforementioned diseases, muscular ependymoma metastasis harbors specific radiological features, such as hypo-anechoic appearance with polylobate margins on ultrasound, nonspecific low SI on T1w and high SI on T2w with variable contrast enhancement on MRI, and being displayed as an heterogenous mass with occasional calcifications on CT scan. Moreover, in the present case hints and distinctive features addressing the diagnosis of paravertebral metastasis of spinal ependymoma were fluid-fluid levels and hemoglobin degradation products in various states of evolution and inhomogeneous post-contrast enhancement. (Table 3) [34]. Of course, the differentiation between bleeding and calcifications can be difficult with MRI and their proper evaluation and differentiation is much easier with CT. Unfortunately, we did not perform a preoperatory CT scan.

Nevertheless, relying on literature data on ependymomas radiological features, we assumed the T1 hyper-intensity to be blood in the context of the altered signal area morphology (fluid-fluid level) [35]. Indeed, hemorrhages and calcification are reported to have a hyper- or hypointense signal in T1 and a low signal in T2 (the low T2 signal at the tumor margins is called cap sign) [36]. Moreover, as the enhancement pattern of MPE is typically homogeneous, its alteration may depend on the hemorrhage. Finally, the susceptibility-weighted imaging (SWI) sequences proved to be able to clearly differentiate between calcification and hemorrhages [37,38]. Histopathological features and genetic and epigenetic features confirm the diagnosis of myxopapillary ependymoma and support the ependymal origin of muscular lesion. Treatment gold standard for symptomatic ependymoma metastasis is complete surgical removal with or without postoperative radiotherapy depending on the extent of disease and the extent of resection [16,39]. Considering all the above, the decision of our spinal and oncologic board was further bolstered by the need of histologically characterize and confirm the nature of the lesion. Even if there are no previous cases suggesting a potential management of paravertebral metastasis of spine ependymoma, our rigorous adherence to the pertinent literature on the treatment strategy of other ependymoma extra neural metastasis granted symptoms relief and no new localization was detected during the last follow up.

## 5. Conclusions

Paravertebral muscle ependymoma metastasis is exceptionally rare. Although anecdotal, they may represent a diagnostic challenge due to multiple differential diagnosis and physicians should be aware of the possibility of their occurrence and the potential treatment strategies.

## Figures and Tables

**Figure 1 brainsci-12-01227-f001:**
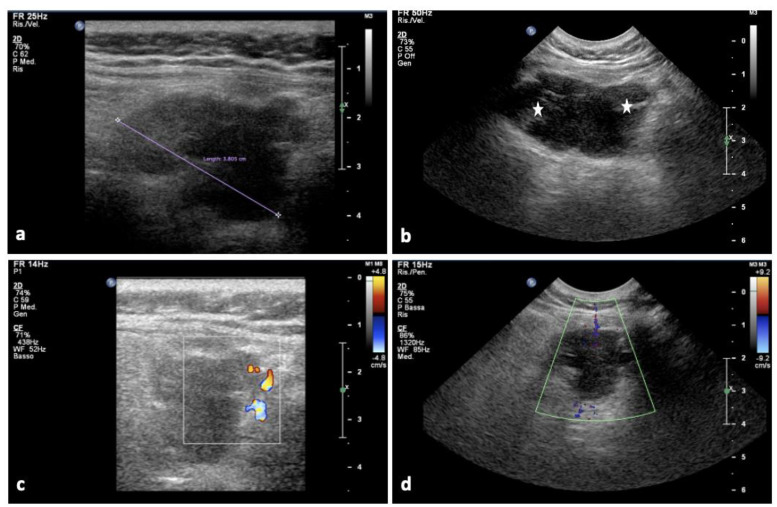
The ultrasound (US) examination demonstrates a hypo-anechoic formation with polylobate margins and a sub-fascial extension to deep left paravertebral muscle site. The lesion has a diameter of 38 mm (**a**) and it is featured by an echogenic heterogeneity due to inner hyper-echoic areas (stars in (**b)**) without vascular signal on color-Doppler and PD (**c**,**d**).

**Figure 2 brainsci-12-01227-f002:**
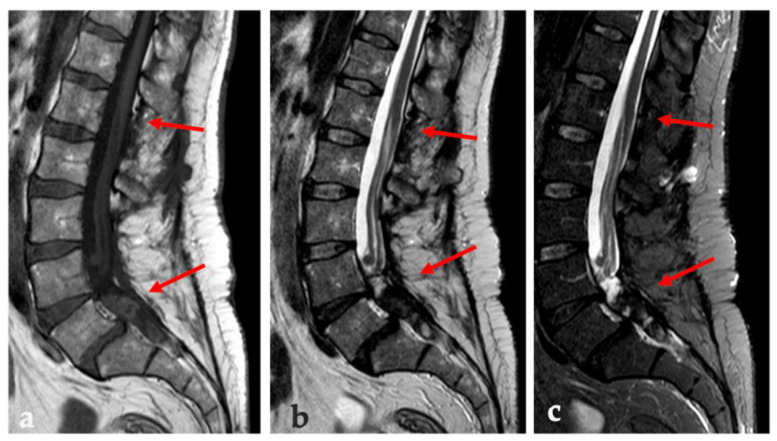
On T1w (**a**), T2w (**b**), STIR (**c**) sequences is shown a thickening of the roots of the cauda with a pseudo-nodular appearance and a tissue component extending from L4 to S2 (the upper and caudal ends of the lesion are pointed by the arrows).

**Figure 3 brainsci-12-01227-f003:**
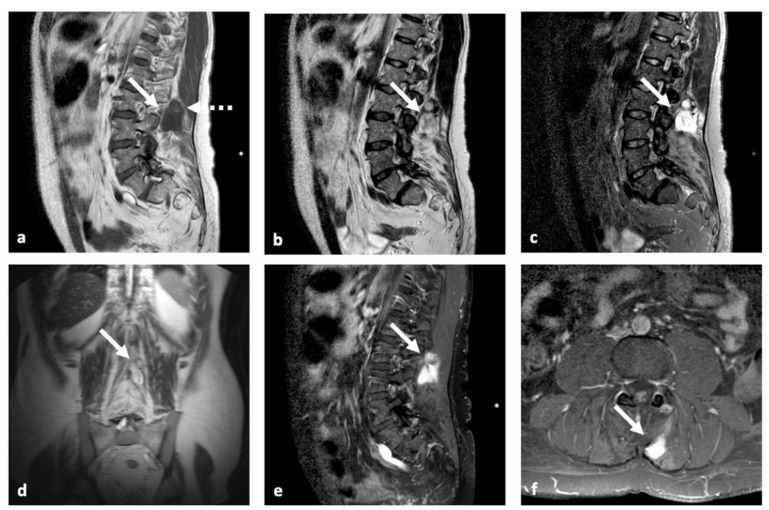
MRI examination—performed by sagittal T1-weight (**a**) and STIR (**c**) sequences, sagittal (**b**) and coronal (**d**) T2-weight, and sagittal (**e**) and axial (**f**) T1-weight contrast-enhanced sequences. The images confirm the presence of a grossly polylobate solid tissue (white arrows in (**a**–**f**)) at level L2–L3 with sub-fascial extension to deep left paravertebral muscle site. This lesion presents a heterogeneous T1-w hypo-intensity with a spontaneous hyper-intensity underlying the presence of a fluid-fluid level (white dotted arrow in (**a**)) due to hemoglobin degradation products. T2w and STIR sequences show heterogeneous hyper-intensity and inhomogeneous but avid post-contrast enhancement. These features are common in MPE metastasis.

**Figure 4 brainsci-12-01227-f004:**
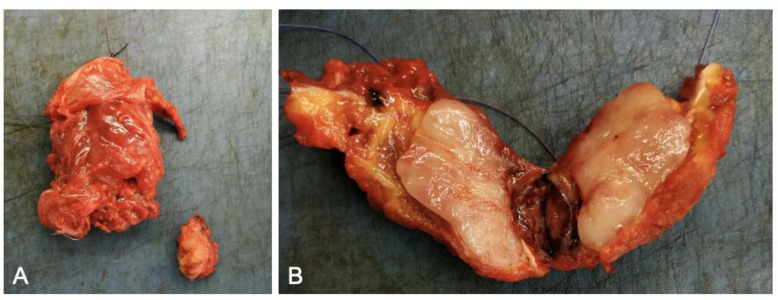
Macroscopic features of the lesion. (**A**) The whole mass encircled by muscle. (**B**) Longitudinal section area.

**Figure 5 brainsci-12-01227-f005:**
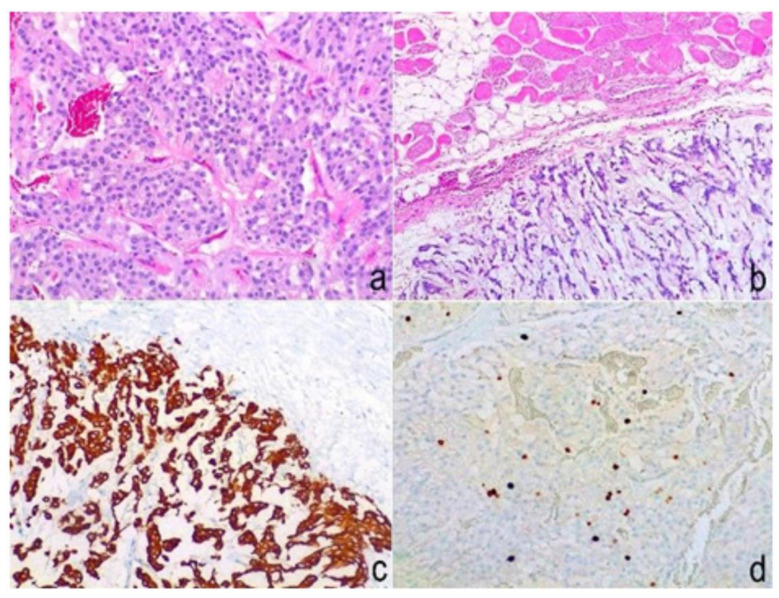
Histopathological images of lesion specimens. (**a**) (Hematoxylin and eosin stain, 200×): the solid component was composed of pseudopapillary structures made of cells with mild cytological atypia, separated by vascular space; (**b**) (hematoxylin and eosin stain, 100×) another component was made by epithelioid cells interspersed within mucoid material and in strict contact with fat tissue and striated muscular fibers; (**c**) neoplastic cells were immunoreactive to GFAP (100×), and (**d**) showed a low (<10%) cellular Ki67 proliferation index (100×).

**Figure 6 brainsci-12-01227-f006:**
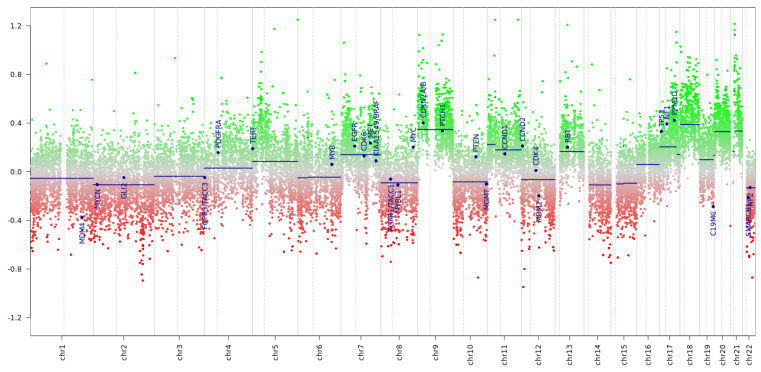
Copy Number Variations (CNV) analyses. Copy Number Variations (CNVs). Analyses of CNVs showed different chromosomal aberrations typical of myxopapillary ependymoma. The dots represent the genes. Genes loss is indicated with red dots, and it is considered significant when inferior to −0.4. Genes gain is indicated with green dots, and it is considered significant when superior to −0.4. Amplification of chromosome 9 was detected, with strong amplification of CDKN2A/B and PTCH1; amplification with gain of materials on chromosomes 7-9-11-13-17-18-20-21.

**Table 1 brainsci-12-01227-t001:** Immunohistochemical analysis on the muscular MPE metastasis. Immunohistochemical markers analyzed in the patient. (EMA, epithelial membrane antigen; GFAP, glial fibrillary acid protein; p, protein).

Immunohistochemical Marker	Result
GFAP	Present
pS100	Present
Cytokeratin-pan	Absent
Cytokeratin 5/6	Absent
EMA	Absent
p40	Absent
Ki67	<10%

**GFAP**: glial fibrillary acidic protein; **p**: protein; **EMA:** epithelial membrane antigen.

**Table 2 brainsci-12-01227-t002:** Cases of extra CNS metastases of ependymomas published in the literature for in last 65 years (since 1955) with localization specification. The latency of extra-axial metastasis occurrence was calculated from the first diagnosis.

Author and Year	Age	Sex	Origin	Histology	Metastases	Latency of Extra-Axial Metastasis Occurrence
Weiss et al., 1955	22	M	CE	MPE	Retroperitoneum, lungs, liver, hilar and tracheobronchial lymph node, mediastinum, IVC, pleura, chest wall, diaphragm, pelvis	10 years
Sharma et al., 1956	29	M	CE	MPE	Lungs, liver, mediastinum, paraaortic lymph node, IVC, pleura	4 years
Peterson et al., 1961	28	F	CE	E	Lungs, liver, vertebrae, paraaortic and tracheobronchial lymph node, mediastinum, IVC, pleura	17 years
Rubinstein et al., 1970	17	F	CE	E	Lung, lymph node, pleura, pelvis, rib, 4° ventricle	29 years
Wight et al., 1973	20	M	CE	MPE	Para aortic lymph node, pleura, pelvis, rib, homerus	38 years
Mavroudis et al., 1977	7	M	CE	E	Lungs	30 years
Morris et al., 1983	10	F	CE/conus	E	Lungs, inguinal lymph node	11 years
Newton et al., 1992	37	M	CE	MPE	Lungs, pleura	<1 year
Newton et al., 1992	16	M	CE	MPE	Abdomen (ventricular-peritoneal shunt)	22 years
Farrier et al., 1994	23	M	TL	E	Vertebrae, bone marrow	12 years
Graf et al., 1999	15	M	CE/conus	MPE	Lungs, liver, abdominal and mediastinal lymph node, pleura, chest wall	40 years
Rickert et al., 1999	55	F	CE	MPE	Lungs	12 years
Vega-Orozco et al., 2011	22	M	CE	MPE	Inguinal lymph node	5 years
Fujimori et al., 2012	28	M	CE	E	Lung, abdominal lymph node, IVC, pelvis	20 years

CE: cauda equina; E: ependymoma; F: female; M: male; MPE: myxopapillary ependymoma; S: sacrum; TL: toraco-lumbar tract.

**Table 3 brainsci-12-01227-t003:** Radiologic features of lesions commonly involving paravertebral muscles.

	Myxopapillary Ependymoma Metastasis	Lipoma	Liposarcoma	Rhabdomyosarcoma	Intramuscular Myxoma
**U/S**	Hypo-anechoic formation with poly lobate margins	Hyperechoic relative to the adjacent muscle	Heterogenous, multi-lobulated, typically well-defined mass of variable ultrasonographic appearance	Non-specific; hypoechoic	Well-defined hypoechoic mass; hyperechoic rim (the bright rim sign)
**MRI**	Nonspecific: low SI on T1w; high SI on T2w; variable C.E.	Identical to fat; no C.E.	Adipose areas; non adipose areas: low SI on T1w, high SI on T2w, variable C.E.	Nonspecific: intermediate SI on both T1w and T2w	Low to intermediate SI on T1w; high SI on T2w; a small rim of fat and edema in adjacent muscles; variable C.E.
**CT**	Heterogenous mass; sometimes calcifications	Identical to fat; no C.E.	Heterogeneous attenuation; thick septa (>2 mm)	Nonspecific: heterogenous mass; calcifications; bone involvement: ill-defined, lytic metastasis	Homogeneous mass with attenuation higher than water and lower than muscles; sometimes mild C.E.

U/S = ultrasound; MRI = magnetic resonance imaging; CT = computer tomography; C.E. = contrast enhancement; T1w, T1-weighted; T2w, T2w-weighted; SI, signal intensity.

## Data Availability

Data are available from the corresponding author upon reasonable request.

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
