# Peer review of "Extra-Neural Metastases of Late Recurrent Myxopapillary Ependymoma to Left Lumbar Paravertebral Muscles: Case Report and Review of the Literature"

_brainsci, 2022, doi:10.3390/brainsci12091227_

Round 1
Reviewer 1 Report
|
The authors report on an extra-neural metastasis of late recurrent myxopapillary ependymoma to the paravertebral muscles. The present case is indeed of great interest, given its rarity. While some case reports are available, a comprehensive clinical and histomolecular characterization of primary tumor and late extra-CNS ependymoma metastasis are absence. Comment 1: It is helpful to discuss the primary tumor and its recurrence (if possible, compare histologic and molecular features as well as surgical and adjuvant treatment). Comment 2: Histological images (figure 4) are in a low resolution. Please take figures in higher resolution (300 dpi). Comment 3: When analyzing the methylation profile of the tumor, the resulting diagnostic score as well as the DKFZ classifier version should be indicated. You report abnormalities on chromosome 9 and, in particular, amplification of the CDKN2A/B and PTCH1 genes. In your opinion, is this the cause of tumorigenesis or tumor metastasis? Distinguish between the use of the terms "amplification" and "gain". Chromosome 11 is also characterized by copy number abnormalities. |
Author Response
Dear Dr. Prof. Dr. Stephen D. Meriney
Editor in Chief
Brain Sciences
We gratefully accepted the comments and the suggestions from the editors and the reviewers. According to them, we have made extensive changes in our manuscript entitled “Extra-neural
metastases of late recurrent myxopapillary ependymoma to left lumbar paravertebral muscles: case
report and review of the literature”. The text as well as its English form have been adapted as per the advices received. All the outlined issues have been addressed, and we hope that in this new form, our manuscript will be suitable for publication on Brain Sciences.
Reviewer 1
The authors report on an extra-neural metastasis of late recurrent myxopapillary ependymoma to the paravertebral muscles. The present case is indeed of great interest, given its rarity. While some case reports are available, a comprehensive clinical and histomolecular characterization of primary tumor and late extra-CNS ependymoma metastasis are absence.
We thank the reviewer for the comments and for the effort employed for improving our manuscript. According to the suggestions, we have made the following changes:
Comment 1: It is helpful to discuss the primary tumor and its recurrence (if possible, compare histologic and molecular features as well as surgical and adjuvant treatment).
- We completed our manuscript by focusing on primary tumor behavior and its risk of recurrence. Moreover, we reported the state of the art in management strategy of the myxopapillary ependymoma accordingly to its local invasiveness and the related difficult to achieve an extensive resection.
Comment 2: Histological images (figure 4) are in a low resolution. Please take figures in higher resolution (300 dpi).
As you suggested, we provided higher resolution images.
Comment 3: When analyzing the methylation profile of the tumor, the resulting diagnostic score as well as the DKFZ classifier version should be indicated. You report abnormalities on chromosome 9 and, in particular, amplification of the CDKN2A/B and PTCH1 genes. In your opinion, is this the cause of tumorigenesis or tumor metastasis? Distinguish between the use of the terms "amplification" and "gain". Chromosome 11 is also characterized by copy number abnormalities.
Many thanks for your comments. We added in the text the alterations involved chromosome 11.
Molecular analyses were carried out only on the second lesion, thus the amplification of the CDKN2A/B and PTCH1 genes, are not sufficient to discriminate between tumorigenesis or metastasis. We have now added a sentence in the results section: Epigenetic and molecular features to specify this point. (Line 130-132).
Reviewer 2 Report
The authors studied a case about Extra-neural metastases of late recurrent myxopapillary ependymoma to left lumbar paravertebral muscles
The paper needs major revision:
Abbreviations must be mentioned from their first appearance ( NSAIDs line 62/ MRI line 69/ PD line 74/ dCC x dAP x dLL line 76/ GFAP line 103/ EMA line 104/ EPIC line 110/ IDAT line 110….).
the subtitle : 3.5. Figures, Tables and Schemes is unnecessary
tables must be readable in themselves: abbreviations must be put at the bottom of the tables
Tables 2 and 3 should not be included in the results section but rather in the discussion
The methods section should be described with more detail.
Figures 1 and 2 are not clear
Put stars and arrows to better visualize the lesion
In the title of figure 2, there are only 3 sequences, whereas there are 4 figures
use a, b, c ….. is better to explain,
the lesion is not as obvious on the chosen MRI sections, preferably add other sections (axial/coronal) or other sequences : T1 Gado
how do you differentiate calcifications from haemorrhage on the MRI sequences? preferably use CT scans
It is preferred to use the latest version of The World Health Organization (WHO) Classification of Tumor of the Central Nervous System ( 2021) as a reference
the majority of paragraphs in the discussion section are without references
Numbers at the beginning of the sentence must be written in letters ( example 14 / line 170 )
The English level is to be improved.
Author Response
Dear Dr. Prof. Dr. Stephen D. Meriney
Editor in Chief
Brain Sciences
We gratefully accepted the comments and the suggestions from the editors and the reviewers. According to them, we have made extensive changes in our manuscript entitled “Extra-neural
metastases of late recurrent myxopapillary ependymoma to left lumbar paravertebral muscles: case
report and review of the literature”. The text as well as its English form have been adapted as per the advices received. All the outlined issues have been addressed, and we hope that in this new form, our manuscript will be suitable for publication on Brain Sciences.
Reviewer 2:
The authors studied a case about Extra-neural metastases of late recurrent myxopapillary ependymoma to left lumbar paravertebral muscles
The paper needs major revision:
As authors, we are very grateful to this reviewer for the comments and suggestion for improving our manuscript. We tried to address all the issues identified. We look forward you may consider our manuscript suitable for publication in its current form.
Abbreviations must be mentioned from their first appearance ( NSAIDs line 62/ MRI line 69/ PD line 74/ dCC x dAP x dLL line 76/ GFAP line 103/ EMA line 104/ EPIC line 110/ IDAT line 110….).
- We carefully checked the manuscript modifying it in order to define abbreviations upon their first appearance. Nevertheless, EPIC and IDAT are not abbreviations, so we left them unchanged in the text.
the subtitle : 3.5. Figures, Tables and Schemes is unnecessary
- As you proposed, we removed the paragraph 3.5 including the figures and the tables in the text.
tables must be readable in themselves: abbreviations must be put at the bottom of the tables
- We followed your suggestion and modified the tables accordingly
Tables 2 and 3 should not be included in the results section but rather in the discussion
- We change the position of the tables placing them in the discussion section, as you interestingly proposed.
The methods section should be described with more detail.
- As you proposed, we markedly revised and rewrite the method section. We look forward that in the current form it may better explain the steps of our clinical study and literature review.
Figures 1 and 2 are not clear
Put stars and arrows to better visualize the lesion
In the title of figure 2, there are only 3 sequences, whereas there are 4 figures
use a, b, c ….. is better to explain,
the lesion is not as obvious on the chosen MRI sections, preferably add other sections (axial/coronal) or other sequences : T1 Gado
- We carefully studied and addressed the issues that you enlighten. We modified the figures via separating the figures showing the paraspinal metastasis (fig.2) and the spinal lesion (fig.3). As well, we added more sequences and the axial sections. Moreover, we included arrows and asterisk to help the reader to clearly identify the lesions.
how do you differentiate calcifications from haemorrhage on the MRI sequences? preferably use CT scans
- Differentiation between bleeding and calcifications can be difficult with MRI and their evaluation is much easier with CT. Unfortunately, we did not perform a preoperatory CT to this patient. Nevertheless, relying on the literature reports of ependymomas radiological features, we are fairly confident that T1 hyper-intensity is attributable to bleeding because of the altered signal area morphology (fluid-fluid level, explained in the figure). Hemorrages and calcification are reported having an hyper- or hypointense signal in T1 and a low signal in T2 (the low T2 signal at the tumor margins is called cap sign). Moreover, the enhancement pattern is typically homogeneous. However, an alteration in the usual homogeneous enhancement pattern in T1 Gado sequences may depend on the hemorrhage.
It is preferred to use the latest version of The World Health Organization (WHO) Classification of Tumor of the Central Nervous System ( 2021) as a reference
- We modified the reference and the text according to your suggestion
the majority of paragraphs in the discussion section are without references
- We included more references in order to bolster the scientific reliability of our paper as you proposed
Numbers at the beginning of the sentence must be written in letters ( example 14 / line 170 )
- We modified the text accordingly
The English level is to be improved
- The manuscript underwent extensive professional English revisions.
Round 2
Reviewer 1 Report
Overall, the authors did a good job addressing the reviewers’ comments and revising the manuscript accordingly.
However, the description of tumor DNA methylation profiling leaves much to be desired. When analyzing the methylation profile of the tumor, the resulting diagnostic score as well as the DKFZ classifier version should be indicated. The authors should familiarize themselves with studies of the genetic profile of MPE. For example, https://doi.org/10.3390/cancers13194954 ; https://doi.org/10.1093/noajnl/vdab043 or https://doi:10.1111/j.1750-3639.2009.00333.x. As you know, numeric whole chromosome changes are frequent in MPE, with gain of chromosome 5, 7, 9, 16 and 18 each in over 50% of cases. In addition, gain of chromosome 11, 13, 17, 20, and 21 identified in this case are highly specific for MPE. The mention of amplification of cancer-suppressor genes such as CDKN2A/B and PTCH1 is confused.
Author Response
Dear Dr. Prof. Dr. Stephen D. Meriney Editor in Chief Brain Sciences We gratefully accepted the comments and the suggestions from the editors and the reviewers, and we further modified accordingly our manuscript entitled “Extra-neural metastases of late recurrent myxopapillary ependymoma to left lumbar paravertebral muscles: case report and review of the literature”. We hope that in this new form, our manuscript will be suitable for publication on Brain Sciences. Reviewer 1: Overall, the authors did a good job addressing the reviewers’ comments and revising the manuscript accordingly. However, the description of tumor DNA methylation profiling leaves much to be desired. When analyzing the methylation profile of the tumor, the resulting diagnostic score as well as the DKFZ classifier version should be indicated. The authors should familiarize themselves with studies of the genetic profile of MPE. For example, https://doi.org/10.3390/cancers13194954 ; https://doi.org/10.1093/noajnl/vdab043 or https://doi:10.1111/j.1750-3639.2009.00333.x. As you know, numeric whole chromosome changes are frequent in MPE, with gain of chromosome 5, 7, 9, 16 and 18 each in over 50% of cases. In addition, gain of chromosome 11, 13, 17, 20, and 21 identified in this case are highly specific for MPE. The mention of amplification of cancer-suppressor genes such as CDKN2A/B and PTCH1 is confused. - Many thanks for suggestions. We have indicated the score of diagnostic methylation analysis (calibrated score:0,97) and the DKFZ methylation classifier utilized (brain classifier_v11b4; version 3.1). (Lines 130-131) According to reviewer suggestions, we have, now, specified in the text the characteristic alterations of the CNV typical of MPEs (lines: 135-136) and we have added the pertinent references (refs:12-13-14 in the revised manuscript). Furthermore, we have deleted the sentence mentioning amplification of CDKN2A/B and PTCH1.Reviewer 2 Report
most of the requested revisions have been made
the figures are clearer
The paper needs minor revision:
the title should be above the table
tables must be readable in themselves: abbreviations must be put at the bottom of the tables
how do you differentiate calcifications from haemorrhage on the MRI sequences? preferably use CT scans
there is two figure 5 ? I think the second one should be eliminated
Author Response
Dear Dr. Prof. Dr. Stephen D. Meriney Editor in Chief Brain Sciences We gratefully accepted the comments and the suggestions from the editors and the reviewers, and we further modified accordingly our manuscript entitled “Extra-neural metastases of late recurrent myxopapillary ependymoma to left lumbar paravertebral muscles: case report and review of the literature”. We hope that in this new form, our manuscript will be suitable for publication on Brain Sciences. Reviewer 2: most of the requested revisions have been made the figures are clearer The paper needs minor revision: - As authors, we are very grateful to this reviewer for accepting our modification and for kindly addressing to us other interesting suggestions aimed at improving out manuscript. We look forward you may consider our manuscript suitable for publication in its current form. the title should be above the table tables must be readable in themselves: abbreviations must be put at the bottom of the tables - We followed your suggestions, and we modified the tables and the position of their title accordingly how do you differentiate calcifications from haemorrhage on the MRI sequences? preferably use CT scans - We addressed this issue in the discussion section: “ Of course, the differentiation between bleeding and calcifications can be difficult with MRI and their proper evaluation and differentiation is much easier with CT. Unfortunately, we did not perform a preoperatory CT scan. Nevertheless, relying on literature data on ependymomas radiological features, we assumed the T1 hyper-intensity to be blood in the context of the altered signal area morphology (fluid-fluid level) [35]. Indeed, hemorrhages and calcification are reported to have a hyper- or hypointense signal in T1 and a low signal in T2 (the low T2 signal at the tumor margins is called cap sign) [36]. Moreover, as the enhancement pattern of MPE is typically homogeneous, its alteration may depend on the hemorrhage. Finally, the susceptibility-weighted imaging (SWI) sequences proved to be able to clearly differentiate between calcification and hemorrhages [37,38].” there is two figure 5 ? I think the second one should be eliminated - We apologize for the mistake. It was a typo. The figure 6 shows the copy number variation analysis, one of the pillars of our clinical and scientifical investigation.